# New Insights into Boron Essentiality in Humans and Animals

**DOI:** 10.3390/ijms23169147

**Published:** 2022-08-15

**Authors:** Andrei Biţă, Ion Romulus Scorei, Tudor Adrian Bălşeanu, Maria Viorica Ciocîlteu, Cornelia Bejenaru, Antonia Radu, Ludovic Everard Bejenaru, Gabriela Rău, George Dan Mogoşanu, Johny Neamţu, Steven A. Benner

**Affiliations:** 1Department of Biochemistry, BioBoron Research Institute, S.C. Natural Research S.R.L., 31B Dunării Street, 207465 Podari, Romania; 2Department of Pharmacognosy & Phytotherapy, Faculty of Pharmacy, University of Medicine and Pharmacy of Craiova, 2 Petru Rareş Street, 200349 Craiova, Romania; 3Department of Physiology, Faculty of Medicine, University of Medicine and Pharmacy of Craiova, 2 Petru Rareş Street, 200349 Craiova, Romania; 4Department of Analytical Chemistry, Faculty of Pharmacy, University of Medicine and Pharmacy of Craiova, 2 Petru Rareş Street, 200349 Craiova, Romania; 5Department of Pharmaceutical Botany, Faculty of Pharmacy, University of Medicine and Pharmacy of Craiova, 2 Petru Rareş Street, 200349 Craiova, Romania; 6Department of Organic Chemistry, Faculty of Pharmacy, University of Medicine and Pharmacy of Craiova, 2 Petru Rareş Street, 200349 Craiova, Romania; 7Department of Physics, Faculty of Pharmacy, University of Medicine and Pharmacy of Craiova, 2 Petru Rareş Street, 200349 Craiova, Romania; 8Foundation for Applied Molecular Evolution (FfAME), 13709 Progress Avenue, Room N112, Alachua, FL 32615, USA

**Keywords:** naturally organic boron-containing compounds, prebiotic candidate, microbiome, intestinal microflora, symbiosis

## Abstract

Boron (B) is considered a prebiotic chemical element with a role in both the origin and evolution of life, as well as an essential micronutrient for some bacteria, plants, fungi, and algae. B has beneficial effects on the biological functions of humans and animals, such as reproduction, growth, calcium metabolism, bone formation, energy metabolism, immunity, and brain function. Naturally organic B (NOB) species may become promising novel prebiotic candidates. NOB-containing compounds have been shown to be essential for the symbiosis between organisms from different kingdoms. New insights into the key role of NOB species in the symbiosis between human/animal hosts and their microbiota will influence the use of natural B-based colon-targeting nutraceuticals. The mechanism of action (MoA) of NOB species is related to the B signaling molecule (autoinducer-2-borate (AI-2B)) as well as the fortification of the colonic mucus gel layer with NOB species from B-rich prebiotic diets. Both the microbiota and the colonic mucus gel layer can become NOB targets. This paper reviews the evidence supporting the essentiality of the NOB species in the symbiosis between the microbiota and the human/animal hosts, with the stated aim of highlighting the MoA and targets of these species.

## 1. Introduction

Boron (B) is considered a prebiotic chemical element with a role in both the origin and evolution of life [1,2,3,4,5], as well as an essential micronutrient (meaning that life cannot be sustained without it) in plants [6,7], some bacteria [8,9], fungi, and algae [10]. Moreover, B is considered beneficial in human and animal nutrition [11,12]. B has not been classified as an essential micronutrient for humans and animals because its biological role has not been clearly identified [13,14,15]. However, B has beneficial effects on the biological functions of humans and animals, such as reproduction, growth, calcium metabolism, bone formation, energy metabolism, immunity, brain function, and steroid hormones, including vitamin D and estrogen [12,16,17,18].

As a scientific term, the word “prebiotic” has two meanings: *(i)* in the origin and evolution of life, “prebiotic” means either the chemical molecules essential for the evolution of life or the raw constituents from which these molecules were formed [3,19]; *(ii)* in nutrition, “prebiotic” today means “a substrate that is selectively utilized by host microorganisms (thereby) conferring a health benefit” [20]. The latest data on the accepted definition of prebiotics from nutrition and regulations have been well reviewed [21]. Currently, prebiotics are based on microbiota-accessible carbohydrates (MACs) [22] but also on other natural phytochemicals, such as polyphenols [23,24], polyunsaturated fatty acids (PUFAs) [25,26], lactose, and lactose derivatives [27]. The effects of prebiotics on human/animal health are increasingly being investigated, with many scientific papers published in this field [28]. Presently, the demonstrated prebiotic effects on health mainly involve benefits for the gastrointestinal (GI) tract, innate and adaptive immune homeostasis, cardiovascular metabolism, mental health, and bone health [29,30,31].

The MACs are some of the most important carbon sources for colon bacteria, as they promote the growth of healthy bacteria and the production of short-chain fatty acids (SCFAs) that can have multiple interactions with the host tissues [32]. A diet that is low in MACs favors bacteria that degrade the mucus and results in the decreased diversity of bacteria with a loss of beneficial bacterial strains. The MACs also contribute to the decreased epithelial integrity of the colon, favoring increased mortality and disease development, as proven within various preclinical and clinical research works [33]. The MAC class also includes some microbiota-accessible naturally organic B (NOB) species, which naturally occur in plants [34] and are present in fruits, vegetables, herbs, nuts, and seeds that are essential to human nutrition [35,36].

Various studies [37,38,39] have shown that sugar-alcohol B esters (SBEs), as NOB species, are crucial in plant development. More recently, SBEs have been proven to be potential modulators of health in humans as well [34,40,41,42]. The identification of the health benefits of SBEs and the further clinical recognition of their importance have become an important direction of research into the science of NOB compounds.

To date, out of all SBEs, only fructoborate (FB) has been clinically tested, proving to have beneficial and quantifiable activity in humans [18,34,35]. The main physiologically stable phyto-SBE compounds are B pectic polysaccharides rhamnogalacturonan II (RG-II) [43], glucose- and fructose-borate esters, *bis*-sucrose esters, and borate polyalcohols containing minimally digestible carbohydrates, such as sorbitol, mannitol, and dulcitol [42,44,45,46], and the recently discovered chlorogenoborate diester complex (CBDC) [47]. In the future, all of these compounds may be promising novel prebiotic candidates [47]. SBEs and CBDC (detected in green coffee beans) are found in numerous vegetables and fruits ingested by humans and animals, and from a biochemical point of view, they are indigestible but microbiota-accessible [34,47].

Recent studies have shown that, since the p*K*_a_ of SBEs is approximately 4.0, their indigestibility is at a maximum above the pH (4.5) of the postprandial stomach [18,34]. In human cells, there are no known biochemical mechanisms or biomolecules that require B, and therefore, B has yet no specific status among nutraceuticals. As the mechanism of action (MoA) of B has not been identified in human/animal cellular metabolism, the market for B nutraceuticals does not yet distinguish between NOB compounds (e.g., SBEs, CBDC) and naturally inorganic B compounds (boric acid (BA), borates).

There is currently a major scientific gap between the use of B compounds in human and animal diets and the MoA of B and its target in the body [12]. However, more recent investigations have shown that in bacteria, the signaling molecule containing B (a furanosyl borate diester or autoinducer-2-borate (AI-2B)) contributes to the animal host’s health (via intestinal flora) and protection against pathogens [18,34,48]. Since SBEs are similar to AI-2B, it has been claimed [34] that SBEs can also increase the beneficial capabilities in some bacteria and reduce virulence in others. Considering the above-mentioned developments, new insights into the essentiality of B species in humans and animals are emerging. This paper reviews the evidence supporting the essentiality of the NOB species in the symbiosis between the microbiota and the human/animal host, with the stated aim of highlighting the MoA and target of these species.

## 2. New Insights into the Essentiality of B Species in Symbiosis across Life Kingdoms

Symbiosis is described as “the close relationship between two organisms of different species, with benefits for one or both individuals” [49]. Symbiosis is any kind of long-term and close biological interaction between two different biological organisms, be it parasitic, commensalic or mutualistic. These close interactions between species are often long term and, for the most part, beneficial to the symbionts [50].

B plays an essential role in several symbioses between plants or animals and bacteria, including the following:

(*i*) *Rhizobium-legume symbioses*: In micromole concentrations, B has been recognized as a key element (essential micronutrient) in the formation of a symbiosis between legumes and nitrogen-fixing bacteria, such as *Rhizobium*, *Azorhizobium* and *Bradyrhizobium* [51,52]. Although, admittedly, B was not found to be essential for these bacteria, these findings show that when B is lacking, the rhizobia-legume “dialogue” is depressed, and the bacterium is identified as a pathogen agent by the plant with disastrous consequences for the symbiosis. The unanimously accepted MoA is that arabinogalactan-protein extensin (AGPE), a hydroxyproline-rich glycoprotein, forms a B complex, while rhizobia cells are separated from it by an exopolysaccharide capsule [8,53]. The existence of a bacterial exopolysaccharide capsule is necessary to prevent the attachment of the AGPE glycoprotein matrix to the surface of the bacterium [54].

(*ii*) *Arbuscular mycorrhizal (AM) symbiosis*: It is well known that a considerable natural accumulation of B in certain plant parts (especially roots and leaves) is controlled by the presence of several mycorrhizal and saprotrophic species [55]. Recently, mycorrhizal symbiosis has been shown to be essential for many plant species to acquire nutrients from soil [56]. Although B does not play a key role in fungi, some of them are carriers for organic B species; consequently, fungal species have been proposed to function as agents for ameliorating B toxicity and for regulating the amount of B species in plants [57]. An interesting fact is that some fungal species accumulate B, while other species exclude it, suggesting that B has specific functions among fungi as well [10]. Recent research has revealed the useful effect of AM symbiosis by diminishing B toxicity in roots and leaves in a citrus rhizome. AM symbiosis improves the tolerance of Carrizo citrus to excessive sources of B by reducing the concentration and toxicity of B in leaves and roots [58]. It is believed that mushrooms do not require B for their own metabolism; the first hypotheses about B in fungi indicated that B could be sequestered as SBEs in forms that are unavailable to fungi. At present, not much information is available about the role of organic B species in AM symbiosis, but future research may demonstrate that the essentiality of B in plants extends beyond the role of B in mycorrhizae/plant symbioses [59].

(*iii*) *Actinorhizal symbiosis*: A nitrogen-fixing bacterium with structure and functionality similar to heterocystous cyanobacteria is *Frankia* spp., an actinomycete symbiont of actinorhizal plants. The following mechanism of action was suggested: B may be necessary for maintaining the envelope of the heterocyst of cyanobacteria [60,61], the envelope of filaments and vesicles of *Frankia* spp. [53], and for activation of the quorum sensing (QS) autoinducer-2 (AI-2) signaling molecule [62]. *Actinomyces* spp. are ubiquitous in the soil and in animal microbiota, including human microbiota [53]. Certain species are commensal in the skin microflora, in the oral flora, intestinal flora and the vaginal flora of humans and animals, and it has been assumed that these microorganisms need B for symbiosis with the host (human/animal) [63].

(*iv*) *Algal-bacterial symbiosis*: B-vibrioferrin (a borate-siderophore complex) was separated from the specific environment of a bacterial “symbiont” from a toxic marine dinoflagellate, although this does not necessarily mean that B plays the role of algal partner [64].

(*v*) *Animal kingdom-bacterial symbiosis*: At present, although there is scarce scientific evidence to show that B deficiency adversely affects the symbiosis between an animal host and its microbiota, studies still provide strong and sufficient arguments for B’s importance in the microbiota’s health, such as the following:

(*a*) In microbiome bacteria, the signaling molecule with B, AI-2B, contributes to the health of the host (via intestinal flora or microbiota) and protection against pathogens [48]. Functionally, AI-2 has been linked to important bacterial processes, such as virulence and biofilm formation, suggesting the possibility of AI-2 reacting with B to produce AI-2B [65]. There is a growing body of work showing that the chemical language of bacteria using the AI-2 signaling molecule is a chemical language both between species and within the same species, as well as across kingdoms (e.g., bacteria and animals) [66,67].

(*b*) Oral administration of BA in toxic concentrations to *Blattella germanica* insects caused dysbiosis of the intestinal microbiota (IM) [68].

(*c*) B is essential for symbiosis between nematodes and mice microbiomes. Low or marginal dietary intake with B may affect the establishment and survival of parasites through its effects on the intestinal microflora [69].

(*d*) Significant induction of *Xenopus laevis* tadpole larval growth has been shown to be correlated with changes in the host’s intestinal microbial communities. These changes in the host’s physiology are due to indirect effects of B that could stimulate bowel maturation, with a beneficial effect on bacteria that promote the host metabolism [70,71].

(*e*) Tartrolon is a natural B compound found in some species of bacteria and has been detected in marine bivalve mollusks of the *Teredinidae* family (a group of saltwater shells found in symbiosis with the *Teredinibacter turnerae* bacterium). Together with the *T. turnerae* bacterium, marine bivalve mollusks achieve cellulose digestion and nitrogen fixation. The antibacterial tartrolon produced by mollusk gill symbionts helps to suppress pathogenic bacteria in the mollusk gut and allows the host to more efficiently absorb the glucose released from the breakdown of lignocellulose [72].

(*f*) For chickens, it was observed that B-based nutrition (as BA) helped to regulate the microbiota following the attack of pathological bacteria. In fact, the BA formed complexes with carbohydrates and phenolic acids from the regular chickens’ diet. A simple calculation shows that 0.1% BA used in the experiment can be totally complexed by fructose and phenolic acids from the diet. The study suggests that B maintains intestinal homeostasis and is effective in controlling *Salmonella enteritidis* infection through the microbiota [73].

(*g*) Dietary B addition proved an influence related to dose on protozoan abundance and rumen microbial fermentation in one-year-old rams. The higher B content in the feces could be explained by its availability for the microbiota; B is not accumulated in any internal organ but only in the intestine. The concentration of B in the rumen fluid has been found to be lower than that in the feces [74].

(*h*) Other animal tests showed that, after five days of feeding the sheep with B, the feces had a high concentration of B (250 ppm), higher than in the urine (140 ppm) [75]. This experiment demonstrated the large capacity of the large intestine to “sequester” B. Additionally, this does not necessarily mean that there is a relationship between B metabolism and the organ volume.

(*i*) It has recently been found that when indigestible B in the nutrition ingested by African ostrich chicks reaches the colon, it interacts with the microbiota, thus stimulating apoptosis and cell proliferation. Here again, the proliferation of the intestinal cells is influenced by the relationship between B and microbiota [76].

(*j*) B in oral washes positively influences periodontal health. BA and calcium fructoborate (CaFB) hydrogels have shown their potential as a treatment option for gingivitis and periodontitis [77]. In another study, the B levels in non-carious teeth were also higher than in carious teeth, and a significant negative correlation was identified only in non-carious teeth group [78].

(*k*) Moderate supplementation with dietary B improved growth performance, the digestibility of crude protein, and the diarrhea index in weaned pigs, regardless of health status [79].

(*l*) In humans, a natural diet rich in B led to an improvement in the oral microbiota and, most importantly, to a decrease in thyroid-stimulating hormone (TSH), which is generally a consequence of dysbiosis [80]. At the same time, the ability to buffer saliva increased significantly after a diet rich in B, and the B level of decayed teeth was lower than that of healthy teeth [81]. Increased B in saliva has a positive effect on dental and oral health and may decrease the formation of cavities and show potential as a treatment option for gingivitis and periodontitis [77]. Important changes in salivary buffering capacity and TSH during a natural B-rich diet are of clinical importance, with dysbiosis being a common finding in thyroid disorders [81]. Recent work has also revealed that B-rich foods result in cardioprotective effects and longevity, improving long-term survival among patients with kidney transplantation (KTR) [77]. Another recent work showed that B-rich foods result in lower mortality risks and a more favorable cardiometabolic risk profile [82].

All these data presented above show that, differing from inorganic B species, B organic species mainly appear to be essential for the symbiosis among organisms from different kingdoms.

## 3. Is AI-2B an Essential Quorum Sensing in the Symbiosis between Microbiota and Hosts?

QS is described as a bacterial communication activity that determines competence, bioluminescence, sporulation, antibiotic production, virulence factor secretion, and biofilm evolution in bacteria [83]. A recent study proved that the breadth of QS includes interkingdom communication, as well as being mediated by many newly determined extracellular signaling molecules called autoinducers (AIs) [84].

To date, two types of AI-2 structures have been reported: (2*S*,4*S*)-2-methyl-2,3,3,4-tetrahydroxytetrahydrofuranborate (*S*-THMF-borate, AI-2B) in *Vibrio harveyi* and (2*R*,4*S*)-2-methyl-2,3,3,4-tetrahydroxytetrahydrofuran (*R*-THMF, AI-2) in *Salmonella typhimurium* [62,85]. AI-2 has been suggested to promote communication between species of bacteria in the mammalian gut [86]. AI-2 also plays an essential role in probiotic functionality and intestinal colonization [87], and is also correlated with gut dysbiosis [88,89]. The AI-2 concentration is correlated with gut microbial disorders, so AI-2 can be considered a new biomarker for dysbiosis [90].

AI-2B has a recognized role in bacterial communication (intra- and inter-species). Recent findings show that cross-communication between kingdoms (i.e., bacteria and eukaryotic cells) takes place via the bacterium AI-2B quorum detection system [91]. The limit of detection for AI-2B can range from 35 nM to 0.4 μM. A larger concentration of AI-2B could not cause luminescence; subsequently, BA hinders AI-2B signaling [85,92]. Nevertheless, at an AI-2B level of 70 μM, luminescence restriction occurred. As shown, AI-2B may modify the constitution of the microbe population from the gut, improve dysbiosis caused by antibiotics and develop a healthy microbiota [93].

There is evidence that B is essential for the development of certain types of bacteria, such as heterocysts of cyanobacteria and actinomycetes of the genus *Frankia* [94]. Additionally, many other bacteria are tolerant to large amounts of B, although its essentiality has not yet been demonstrated for them. For example, 15 strains of bacteria (*Algoriphagus*, *Arthrobacter*, *Bacillus*, *Gracilibacillus*, *Lysinibacillus*, and *Rhodococcus* taxa) were isolated and then shown to have tolerance to high concentrations of B [95,96].

In symbioses across kingdoms, bacteria use the ability of B to attach to glycoproteins, thus blocking the bacterium from infecting the host of the symbiosis. In addition, the discovery of bacteria containing one B-signaling molecule (an AI-2 self-inducer, identified as furanosyl borate diester) revealed a surprising role for B in detecting the bacterial quorum. AI-2B is a new signaling molecule that serves as a universal bacterial signal for communication between species and between kingdoms [97].

IM is distributed across the whole GI tract in a heterogeneous way. Collectively, in humans, it is an ecosystem typically weighing 1.5 kg, being formed of more than 1500 bacteria and more than 1000 other species (for instance, viruses, fungi, parasites, phages, and archaebacteria) [98,99]. The most important bacterial phyla of a healthy IM are *Firmicutes* and *Bacteroidetes*, followed by *Fusobacteria*, *Actinobacteria*, and *Proteobacteria*. Since the most important species are *Faecalibacterium*, *Bacteroides*, and *Bifidobacterium*, IM achieves several functions, such as maintaining metabolic homeostasis, nutrient absorption, defense against infections, and the development of mucosa and systemic immunity. AI-2B could influence commensal bacterial behaviors to maintain the balance between *Bacteroidetes* and *Firmicutes* species. AI-2B is generated by multiple bacterial phyla found in the GI tract, such as *Bacteroides* spp., *Eubacterium rectale*, *Ruminococcus* spp., and *Lactobacillus* spp. [48,100].

These remarks support the hypothesis that AI-2B is an essential signaling molecule that might regulate bacteria, community dynamics, and behavior in the microbiota, and could also modulate the composition of the microbiota under dysbiosis conditions. AI-2B production by one phylum may affect the expression of genes of other species and can promote communication between species, allowing bacteria to change their behavior, namely their virulence, luminescence, and biofilm formation, between different species [65,101,102]. This feature makes AI-2B a great candidate for modulating interactions between cells in mammalian intestines, where thousands of bacterial phyla coexist and communicate [103]. An example of the protective role of commensal microbes against pathogenic bacteria is as follows: AI-2B produced from *Ruminococcus obeum* can confuse *Vibrio cholerae*, resulting in the premature repression of AI-2B QS-mediated virulence and decreased colonization in the intestine [104].

## 4. Are NOB Species Essential for Healthy Human/Animal Microbiome Symbiosis?

Nutritional essentiality was clearly established as a concept approximately 100 years ago. This resulted from the observation that certain pathologies may be stopped by including a specific food in the diet. Physiological essentiality and nutritional essentiality are two different concepts. Physiological essentiality means an indispensable material for life, while nutritional essentiality means an indispensable material in the diet [105]. A substance is usually considered to be nutritionally essential if a substance deficiency from the diet results in biological dysfunction. The higher intake of that substance prevents biological dysfunction or makes it reversible. Below, we attempt to prove the nutritional essentiality of NOB species based on standard criteria of nutritional essentiality [105]:

(*i*) “*The substance is necessary in the diet for growth, health and survival*”. B species, such as carbohydrate and phenolic acids esters, are needed for healthy symbiosis between human/animal hosts and the microbiota. The literature holds evidence that certain bacteria need B (AI-2B), a B carbohydrate, for communication, and our preliminary data show that the colonic mucus gel layer of rats also contains B (Figure 1). Our recent discovery of the presence of B within the colonic mucus will need to be confirmed by other laboratories. This notwithstanding, glycoproteins containing B are used in all symbiosis processes between kingdoms in which B is involved and in controlling the interaction between the two entities taking part in the symbiosis process [52].

(*ii*) “*Its absence from the diet or inadequate intake results in characteristic signs of a deficit disease and, ultimately, death*”. The absence of B species from the diet could be a determinant, in our hypothesis, of the permeability of the intestine (“leaky gut syndrome”) and dysbiosis. Dysbiosis and permeability of the intestine cause serious pathologies that lead to death, including inflammation, cancer, heart and brain diseases, arthritis, and bone diseases [108,109]. In most diseases, the composition of the microbiota is altered, causing pathophysiological diseases in vital human organs. The complex interactions between the IM and the host immune system influence the body’s functions, resulting in the formation of an “axis” between the IM and various organs [110]. The “host-microbe” metabolic axis provides systemic multidirectional communication between the cellular pathways of the host and different microbial phyla in the microbiota. Within these axes, various microbes sequentially regulate metabolic reactions, producing bile acids, choline, and SCFAs, which are vital to human/animal health. These metabolites contribute to the metabolic phenotype of the host and the risk of developing the disease [111]. In our view, B deficiency from the symbiosis process may lead to dysbiosis and gut permeability by decreasing the concentration of AI-2B. At the same time, decreasing the concentration of B in the colonic mucus induces the formation of bacterial metabolites and proinflammatory cytokines, which could help the pathophysiological mechanisms of osteoarthritis (OA) [112,113,114]. This mechanism could explain why in areas with a high content of B, people have a low OA index [115]. In our opinion, in the future, B analysis from the feces and colonic mucus could become an important marker indicating the lack of B from nutrition and a predictive factor for several diseases resulting from unhealthy symbiosis. Subsequently, the commensal bacteria need B as an essential component of their diet for survival, thus proving B’s role in modulating microbiome physiology [116]. Furthermore, the male microbiome is different from the female one [117]. This means that males need a different amount of dietary B than females. Recent studies have shown that there are differences in the level of B in the hair of men and women, with women having almost two times less B in their hair [118]. Thus, women are more susceptible to diseases that are caused by B deficiency in their diet. Subsequently monitoring the AI-2-borate marker in feces will open new scientific horizons for monitoring the prophylaxis of diseases due to a B deficiency in food (i.e., OA, osteoporosis, ovarian cancer, polycystic ovary syndrome, obesity, inflammatory bowel disease, diabetes, fatty liver disease, allergic diseases, and cardiovascular diseases) [109,119,120]. In the future, studies aiming to find the causal relationship among sex, the microbiota, and disease will be crucial. The alteration of the IM from normobiosis to dysbiosis may also be different in women and men. Moreover, different diet-induced changes to the gut microbiota by gender suggest that women and men could differentially benefit from the consumption of a specific B diet, depending on their gender and disease.

(*iii*) “*Growth failure and/or characteristic signs of deficiency are prevented only by the nutrient or a specific precursor of it, not by other substances*”. NOB species are prebiotic candidates that may help with bacterial communication via AI-2B [34] and strengthen the mucus gel lining for the human colon (Figure 1). Consequently, the effects of B species deficiency in the microbiota could be: (*a*) dysbiosis, an alteration in the symbiosis between the human/animal host and the microbiota; in our opinion, this is due to the lack of B, which results in the deficiency of the AI-2B signaling molecule [18,34]; (*b*) increased intestinal permeability (known as “leaky gut syndrome”) and translocation of the IM from the gut lumen to the systemic circulation due to the lack of B in the structure of the mucin gel. From mucin separation techniques, the interaction of B with glycosylation sites within mucins and O-glycosylated linear glycoproteins is well known [121]. B deficiency thus determines the interaction of the bacterial biofilm directly with the membranes of the host cell and, therefore, the infection of the host. Since 2008, B-stabilized glycoproteins have been claimed to be important for signaling during symbiosis in plants [122]. The description of MoA for B-stabilized glycoproteins has created the probability that identical membrane constituents in animal cells may have a similar role due to B [122]. In all symbiosis processes in which B is essential, this cannot be replaced by any other supporting symbiosis [6].

(*iv*) “*Below some critical level of intake of the nutrient, growth response and/or severity of signs of deficiency are proportional to the amount consumed*”. There is a direct correlation between the amount of B species ingested, the amount of B in feces [75,123,124] and the level of B in the gel layer of colonic mucus following NOB-supplemented diet in rats (Figure 1). It is known that in OA, diet modification by increasing the B intake results in diminishing the OA’s severity, correlated with the amount of ingested B [115,125].

(*v*) “*The substance (NOB compounds) is not synthesized in the body and is therefore required to be obtained from the diet for some critical function throughout life*”. Humans and other animal cells do not have a metabolic pathway to synthesize organic B compounds, so these must be obtained from plants. Although the World Health Organization (WHO) has still not accepted B as an essential element for humans with a key role in the human metabolism, this may change as our insights are confirmed by other laboratories. Furthermore, confirmation would vindicate Newnham’s 28-year-old recommendation to use B for preventing and managing arthritis and osteoporosis [115].

## 5. Are NOB Species Novel Prebiotic Candidates?

When discussing B’s essentiality, we must consider not the element itself but the molecular species that incorporate it [18]. A few such essential B species have been detected in bacteria (AI-2B), in fungi (SBEs), and in higher plants (SBEs). Recently, a CBDC chemical form was discovered in green coffee beans [47].

Essentiality should be associated with one specific kind of speciation for the same element. In our view, NOB complexes (e.g., the B-RG-II complex, SBEs, organic polyhydroxy acid borate esters, *bis*-sucrose borate complexes, amino acid borate esters and, recently, CBDC) may be prebiotic candidates in human/animal nutrition. This is different from BA/borates, which cannot be prebiotic compounds, as these inorganic compounds are digestible and toxic to microorganisms [34]. In general, in the acidic gastric medium, soluble and insoluble organic B species degrade into B monoesters and diesters [18,34]. As natural B monoesters and diesters (polyalcohols, organic acids sugars, and CBDC) have a p*K*_a_ of between 2.5 and 5.0, many of them do not degrade and therefore remain mainly in the organic form of B. At the same time, because BA can be associated with *cis*-diol organic compounds (fructose, ribose, glucose, and phenolic acids), in the superior part of the digestive tract, the NOB species is probably reconstituted even at pH 4.5 postprandially [18].

Our 30-year research has attempted to fill this scientific gap by studying the many chemical and biochemical reactions in which B is involved [3,4,18,34,35,41,77,126,127,128,129,130,131,132,133,134]. Recently, we analyzed some of our old animal studies and found that at the beginning of the B diet, the increase in the B concentration in feces could be explained by NOB species’ indigestibility; then, NOB is released into the colon, where the microbiota needs it [135,136,137,138]. We evaluated in vivo the ability of a protein concentrate additive enriched in NOB (as FB) to diminish the toxic effect of a corn-based diet contaminated with *Fusarium* toxins in piglets. FB has been hypothesized to have stimulated the activity of intestinal microflora, knowing that mycotoxins can disrupt the IM by altering the relative abundance of the commensal bacteria [139,140]. The natural explanation hypothesis for these results was that the human/animal body did not need B, but the IM needed B to create a healthy symbiotic relationship with the host. Subsequently, the MoA of NOB species is as follows: supplying NOB species to bacteria that require B to communicate (AI-2B), and the fortification of the colonic mucus gel layer with organic B esters (this is similar to plant symbiosis, to protect the host from bacterial infection). Consequently, NOB compounds are both a source of B (essential for symbiosis) and a source of carbon for the specific nutrition of microbiota.

Given the above-mentioned MoA, both the microbiota and the gel layer of the colonic mucus become targets for NOB species. Therefore, in the future, NOB species may become promising novel prebiotic candidates. For instance, in the acidic gastric environment, NOB species (such as FB) have a p*K*_a_ of approximately 4.0, so they do not dissociate in the postprandial upper gastric system (Figure 2; Appendix A) [34,141,142,143,144,145,146]. In addition, indirect evidence of B’s presence in the gel layer of colonic mucus is the probable result of B “sequestration” in the colon during B nutrition in animals [75].

## 6. Perspectives

New discoveries about metabolism and molecular nutrition have increased the level of our understanding in the relationship between the colon and human health. As this understanding has improved, the colon has come to be recognized as the optimum location to improve the bioavailability of food due to its distinct features, including its nearly neutral pH, its low enzymatic activity, and its long transit time [147]. Ideally, nutraceuticals should be preserved in the aggressive medium of the superior GI tract and then released into the colon to achieve a successful distribution in the colon and to obtain their full bioavailability [148].

Establishing the essentiality of NOB species for a healthy relationship between microbiota and their human/animal hosts will mark a new approach to the use of these species as prebiotic candidates in human and animal nutrition. The literature explains that the health of humans/animal organisms depends on the health of the microbiota [116,149], supporting the view that the role of NOB species is crucial for the health of the human/animal organism [34]. This role has resulted from many scientific experiments shown that NOB species increase the buffering capacity of saliva [77,81], have a positive impact on the intestinal and oral microbiome [76], protect important probiotic bacteria, namely *Bifidobacterium* spp. and *Lactobacillus* spp. [150,151,152], improve SCFAs production [74], are essential for improving the integrity and impermeability of the intestinal barrier [76], and improve the immunity and anti-inflammatory and antioxidant actions of the microbiota [153]. The microbiota influences the health of the body through the “axes” already formulated and studied in the literature: the “gut-brain” axis, “gut-immunity” axis, “gut-bone” axis, “gut-cartilage” axis, “gut-heart” axis, “gut-lung” axis, and “gut-thyroid” axis [111]. This may explain the beneficial role of B in preventing certain diseases, such as OA [115], osteoporosis [154], rheumatoid arthritis [155], cardiovascular inflammation [18,131,156], depression [157], obesity [158], diabetes [159], viral and bacterial infections [133,160], and thyroid diseases [161].

## 7. Conclusions

New insights into the essentiality of B for humans and animals suggest that a new hypothesis might be required to understand the role of B in health: “Naturally occurring B species are essential for human/animal host healthy microbiome symbiosis”. This, in turn, will drive new insights into the essentiality of NOB species for healthy symbiosis between the human/animal host and the microbiota, which in turn will guide the use of natural B-based nutraceuticals to target the colon as “colonic foods”. The MoA of NOB species may be primarily related to B molecule signaling (AI-2B), but may also fortify the colonic mucus with B from the specific prebiotic diet.

Subsequently, the key aspects of NOB species’ use in nutrition are:

(*i*) NOB species are novel and potential prebiotic candidates.

(*ii*) NOB species are needed for the symbiosis between bacteria and human/animal hosts.

(*iii*) NOB species can be considered effective novel prebiotics because more than 95% of them reach the colon and do not dissociate in the form of inorganic B (Figure 2; Appendix A).

(*iv*) BA, borax and all inorganic B salts are not prebiotics because they are digestible, leading to B dissociation and have shown cytotoxic and genotoxic activity for microbiota [162,163,164]. The BA is highly available in the bloodstream, while NOB species are indigestible and therefore reach the colon.

(*v*) NOB species are likely to function as carriers for (*a*) carbon to support microbiome growth and (*b*) the essential B element needed for healthy symbiosis.

From a practical point of view, the essentiality of NOB species will open up new opportunities for supplementing B in human/animal nutrition to stay healthy and live long [18,34,82,165]. The dietary intake of NOB molecules could have an important role in the life extension of humans [18]. This is because NOB species intakes have been shown to moderate or alleviate several pathological conditions associated with aging, including cancer, cognitive decline, sarcopenia, and bone health [34]. These findings indicate that a diet rich in B will promote a healthy life expectancy [14,18,34].

New knowledge about the essentiality of NOB species for a healthy symbiosis between human/animal hosts and microbiota will lead to the use of natural B-based nutraceuticals to target the human/animal microbiome (gut, oral, vaginal, skin, and scalp microbiomes). Of these, the gut microbiome is the most important for human health. Subsequently, NOB species have become novel prebiotic candidates and target the colon as novel colonic foods. Moreover, NOB species target colon nutrition, resulting in a healthy gut microbiome, as well as a healthy microbiome in the mouth, vagina, skin, and scalp.

In conclusion, NOB species are novel prebiotic candidates (“a prebiotic is a substrate that is selectively utilized by host microorganisms conferring a health benefit”) [21]; they are indigestible, and they do not dissociate postprandially at pH 4.5 in the upper gastric system (Figure 2). An increasing number of studies show that nutrition may influence gut microbiota and that human health is crucially dependent on the healthy microbiota [166]. Prospective nutrition will be personalized depending on microbiota type for every human being [167]. Thus, NOB species will become essential for personalized nutrition as further novel prebiotic microbiota-accessible candidates. In the future, the colon’s ability to absorb B in the mucus gel layer may be a new target for the B neutron capture therapy (BNCT) of colon cancers by using ^10^B-enriched prebiotic NOB compounds [138,168,169]. Moreover, supplements containing naturally occurring B should be taken preventively at any time because these supplements ensure the capture and extinction of increased radiation from the living environment. Therefore, these supplements are effective as chemical protection against radiation in conditions of natural radioactivity in soil and water and/or nuclear disasters (nuclear accidents or attacks with nuclear bombs) [170,171,172]. Recently, the IM and its associated metabolites have been proven to play a major role in high-dose radiation protection [173]. Researchers have noticed that only a group of mice resistant to radiation had two main important classes of bacteria in their gut: *Lachnospiraceae* and *Enterococcaceae*. These bacteria need B for the biosynthesis of AI-2B, and the MoA that we proposed for NOB complexes could explain why these complexes ensure absorption of neutrons and gamma-radiation when they reach the colon [138,168,169]. Additionally, in areas where the natural radiation of the soil is increased, the concentration of B in the blood of animals and humans is very low because B in the body is consumed by natural radiation from the environment and transformed into lithium and helium, but only the ^10^B isotope conducts the reaction [174,175,176]. In these areas, people have very advanced OA (over 70%) [177,178,179,180].

## Figures and Tables

**Figure 1 ijms-23-09147-f001:**
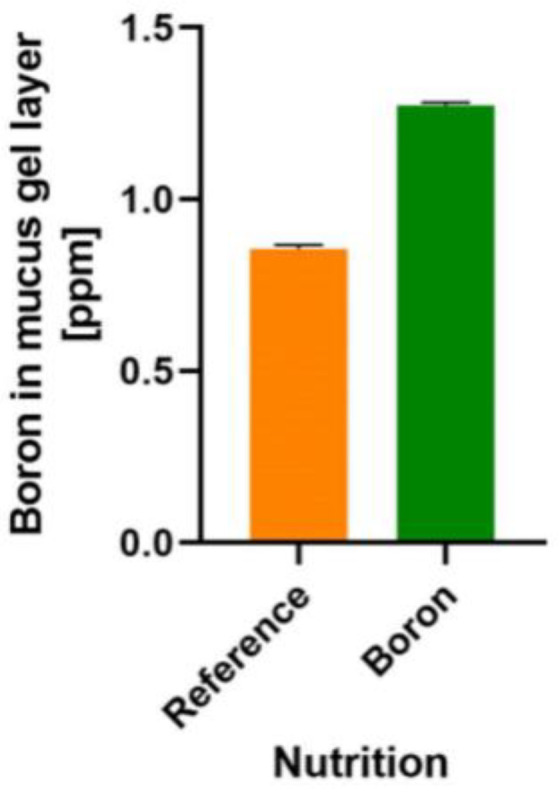
Boron detection in a rat’s colonic mucus gel layer. Reference: normal diet [106]. Boron: CaFB (NOB) supplemented diet. Boron detection was performed by UHPLC/MS [107] (Part S1). CaFB: calcium fructoborate; NOB: naturally organic boron; UHPLC/MS: ultra-high-performance liquid chromatography/mass spectrometry.

**Figure 2 ijms-23-09147-f002:**
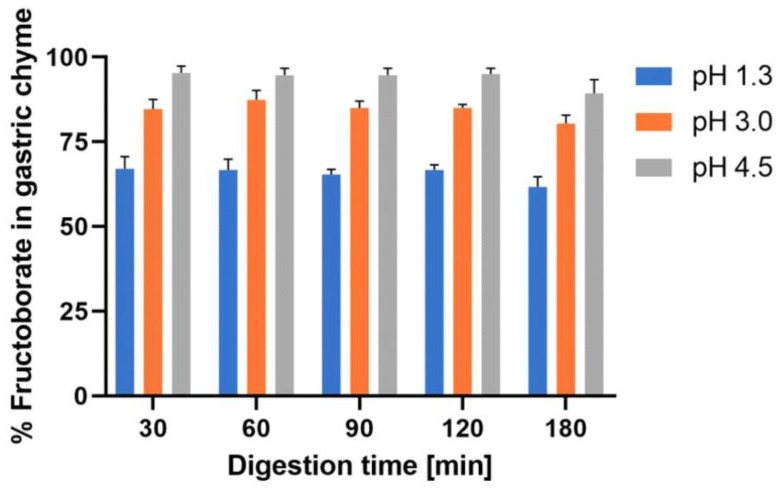
In vitro simulation of gastric digestion of NOB species (Part S2 [142,143,144,145,146]; Appendix A). NOB: Naturally organic boron.

## Data Availability

Data described in the manuscript will be made publicly and freely available without restriction at https://docs.google.com/document/d/1LXqh6cYozOZSCsq9R4TyTBqv5lawr-ZG/edit?usp=sharing&ouid=110601923775251692590&rtpof=true&sd=true (accessed on 15 April 2022).

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
