# Peer review of "New Insights into Boron Essentiality in Humans and Animals"

_ijms, 2022, doi:10.3390/ijms23169147_

Round 1

Reviewer 1 Report

In this article, the authors reviewed the boron essentiality of the natural organic boron species in humans and animals. In concluding part, the authors mentioned the boron in the body is consumed by natural radiation, but only boron-10 isotope conduct the reaction, to make it clear, the following review papers need to be involved: Molecules 2020, 25, 828; Molecules 2021, 26, 3309; and <<Fundamentals and applications of boron chemistry>> (1st Edition-March 17, 2022, eBook ISBN: 9780128221266).

Author Response

Answer to Reviewer 1

Dear Reviewer,

First of all, we would like to address many thanks for your accurate observations and valuable comments. We used all these and improved the paper accordingly.

All changes in the revised manuscript have been marked up using the “Track Changes” function of MS Word.

The following changes have been made for the manuscript # ijms-1823150:

Reviewer’s question/comment:

In concluding part, the authors mentioned the boron in the body is consumed by natural radiation, but only boron-10 isotope conduct the reaction, to make it clear, the following review papers need to be involved: Molecules 2020, 25, 828; Molecules 2021, 26, 3309; and <<Fundamentals and applications of boron chemistry>> (1st Edition-March 17, 2022, eBook ISBN: 9780128221266).

Answer:

In concluding part, we added the following sentence: “… but only the 10B isotope conducts the reaction [174–176].” (p. 11, lines 1025 & 1026), with the appropriate references: Ali et al. (2020) [174], Coghi et al. (2021) [175] and Zhu & Hosmane (2022) [176] (p. 20, lines 1592–1598).

Kind regards,

Ion Romulus SCOREI, Professor, PhD

Reviewer 2 Report

The purpose of this study is to examine the data that supports the essentiality of the NOB species in the symbiosis between the microbiota and the human/animal hosts. The stated objective of this paper is to emphasize the mechanism of action and target of these species. Authors described B playing an essential role in symbiosis, with 5 symbiosis examples. It was well written, with good organization, and good logical structure.

I have a few questions:

Page 3, (iii) Actinorhizal symbiosis

It was too short, it did not describe the MoA.

Page 3, (iv)Algalbacterial symbiosis

Separated B–vibrioferrin, is not necessary meaning NoB is playing the role for algal partner.

Page 4, “There is a growing body of work showing that the “language” of bacteria using AI-2 with B is a “language” both between species and within the same species, as well as across kingdoms (e.g., bacteria and animals)”

Is the “language” here could be understood as a signal?

Page 4, “Other animal tests showed that, after five days of feeding the sheep with B, the feces had a high concentration of B (250 ppm), higher than in the urine (140 ppm). This experiment demonstrated the large capacity of the large intestine to “sequester” B.”

By reference 72, it was pointed out that the ingestion and excretion of B were balanced 5 days after the beginning of the B dosing.

This is not necessarily mean there is much relation between B metabolism the organ volume.

Page 4, “B in oral washes positively influences periodontal health. But the next sentence, B addition did not influence the dentin and enamel of rabbits given high-energy nourishment.”

This is not a supporting statement.

Author Response

Answer to Reviewer 2

Dear Reviewer,

First of all, we would like to address many thanks for your accurate observations and valuable comments. We used all these and improved the paper accordingly.

All changes in the revised manuscript have been marked up using the “Track Changes” function of MS Word.

The following changes have been made for the manuscript # ijms-1823150:

Reviewer’s question/comment:

Page 3, (iii) Actinorhizal symbiosis

It was too short, it did not describe the MoA.

Answer:

The following paragraph has been inserted into the manuscript (p. 3, lines 163–166): “The following mechanism of action was suggested: B may be necessary for maintaining the envelope of the heterocyst of cyanobacteria [60,61], the envelope of filaments and vesicles of Frankia spp. [53], and for activation of the quorum sensing (QS) autoinducer-2 (AI-2) signaling molecule [62].”, with the appropriate references: Bolaños et al. (2002) [53], Bonilla et al. (1990) [60], Garcia-Gonzalez et al. (1991) [61] and Chen et al. (2002) [62] (p. 14, lines 1229–1231, 1249–1256).

Reviewer’s question/comment:

Page 3, (iv) Algal–bacterial symbiosis

Separated B–vibrioferrin, is not necessary meaning NoB is playing the role for algal partner.

Answer:

The sentence: “… indicating that B has a possible biological function for the symbiosis of the algal partner [61].” has been removed.

The following sentence has been inserted in the manuscript (p. 4, lines 184 & 185): “… although this does not necessarily mean that B plays the role of algal partner [64].”, with the appropriate reference: Harris et al. (2007) [64] (p. 14, lines 1260 & 1261).

Reviewer’s question/comment:

Page 4, “There is a growing body of work showing that the “language” of bacteria using AI-2 with B is a “language” both between species and within the same species, as well as across kingdoms (e.g., bacteria and animals)”.

Is the “language” here could be understood as a signal?

Answer:

The paragraph has been rephrased for a better understanding (p. 4, lines 194–196): “There is a growing body of work showing that the chemical language of bacteria using the AI-2 signaling molecule is a chemical language both between species and within the same species, as well as across kingdoms (e.g., bacteria and animals) [66,67].”, with the appropriate references: Kaper & Sperandio (2005) [66] and Pereira et al. (2013) [67] (p. 14, lines 1265–1268).

Reviewer’s question/comment:

Page 4, “Other animal tests showed that, after five days of feeding the sheep with B, the feces had a high concentration of B (250 ppm), higher than in the urine (140 ppm). This experiment demonstrated the large capacity of the large intestine to “sequester” B.”

By reference 72, it was pointed out that the ingestion and excretion of B were balanced 5 days after the beginning of the B dosing.

This is not necessarily mean there is much relation between B metabolism the organ volume.

Answer:

The following sentence has been inserted (p. 4, lines 227 & 228): “Additionally, this does not necessarily mean that there is a relationship between B metabolism and the organ volume.” The new reference [75] (old reference [72]) has been placed where appropriate (p. 4, line 225): Miyamoto et al. [75] (p. 15, lines 1292–1294).

Reviewer’s question/comment:

Page 4, “B in oral washes positively influences periodontal health. But the next sentence, B addition did not influence the dentin and enamel of rabbits given high-energy nourishment.”

This is not a supporting statement.

Answer:

The sentence: “During a specific experiment in which B was administered by gavage, B addition did not influence the dentin and enamel of rabbits given a high-energy nourishment [74].” has been removed, as well as the old reference [74] (Hakki et al., 2021).

The following sentence has been inserted (p. 4, lines 233–237): “BA and calcium fructoborate (CaFB) hydrogels showed their potential as a treatment option for gingivitis and periodontitis [77]. Furthermore, the B levels in non-carious
teeth were also higher than in carious teeth, and significant negative correlation was identified only in non-carious teeth group [78].”, with the appropriate references: MitruÅ£ et al. (2021) [77] and Kuru et al. (2020) [78] (p. 15, lines 1298–1303).

Reviewer’s question/comment:

English language and style are fine/minor spell check required.

Answer:

English language editing for the revised manuscript has been made by MDPI Author Services (Certificate 48855/August 11, 2022).

Kind regards,

Ion Romulus SCOREI, Professor, PhD
